# PD-L1 Expression in Neoplastic and Immune Cells of Thymic Epithelial Tumors: Correlations with Disease Characteristics and HDAC Expression

**DOI:** 10.3390/biomedicines12040772

**Published:** 2024-03-31

**Authors:** Ioanna E. Stergiou, Kostas Palamaris, Georgia Levidou, Maria Tzimou, Stavros P. Papadakos, Georgios Mandrakis, Christos Masaoutis, Dimitra Rontogianni, Stamatios Theocharis

**Affiliations:** 1Department of Pathophysiology, School of Medicine, National and Kapodistrian University of Athens, 11527 Athens, Greece; stergiouioa@med.uoa.gr; 2First Department of Pathology, School of Medicine, National and Kapodistrian University of Athens, 10679 Athens, Greece; kpalamaris@yahoo.gr (K.P.); tzimoumarialuisa@gmail.com (M.T.); stpap@med.uoa.gr (S.P.P.); giormandr@biol.uoa.gr (G.M.); cmasaout@med.uoa.gr (C.M.); 3Second Department of Pathology, Paracelsus Medical University, 90419 Nurenberg, Germany; 4Department of Pathology, Evangelismos General Hospital of Athens, 10676 Athens, Greece; dgian@otenet.gr

**Keywords:** thymic epithelial tumors, thymoma, PD-L1, HDAC

## Abstract

Background: Programmed death-ligand 1 (PD-L1) expression in neoplastic and immune cells of the tumor microenvironment determines the efficacy of antitumor immunity, while it can be regulated at the epigenetic level by various factors, including HDACs. In this study, we aim to evaluate the expression patterns of PD-L1 in thymic epithelial tumors (TETs), while we attempt the first correlation analysis between PD-L1 and histone deacetylases (HDACs) expression. Methods: Immunohistochemistry was used to evaluate the expression of PD-L1 in tumor and immune cells of 91 TETs with SP263 and SP142 antibody clones, as well as the expressions of HDCA1, -2, -3, -4, -5, and -6. Results: The PD-L1 tumor proportion score (TPS) was higher, while the immune cell score (IC-score) was lower in the more aggressive TET subtypes and in more advanced Masaoka–Koga stages. A positive correlation between PD-L1 and HDAC-3, -4, and -5 cytoplasmic expression was identified. Conclusions: Higher PD-L1 expression in neoplastic cells and lower PD-L1 expression in immune cells of TETs characterizes more aggressive and advanced neoplasms. Correlations between PD-L1 and HDAC expression unravel the impact of epigenetic regulation on the expression of immune checkpoint molecules in TETs, with possible future applications in combined therapeutic targeting.

## 1. Introduction

Thymic epithelial tumors (TETs) are the most common mediastinal neoplasms, presenting clinicopathological and molecular heterogeneity. The most recent WHO classification divides them into six major histological subtypes, characterized by progressively increasing malignant behavior and worsening prognosis: A, AB, B1, B2, B3, C [1]. The distinctive features of these subtypes are defined both by neoplastic cells and tumor microenvironment (TME). Tumors of the A subtype consist foremost of spindle/oval epithelial cells, with only scarce lymphocytes, while the B1, B2, and B3 subtypes comprise a combination of neoplastic epithelioid cells with progressive aggravation of atypia and diminishment of the lymphocytic population (B1 = lymphocyte-rich to B3 = lymphocyte-poor). Finally, tumors of the C subtype are defined as thymic carcinomas (TCs) and display morphological and immunohistochemical traits of distinct malignancies, mainly of squamous differentiation. From a molecular perspective, there is significant overlap between the different histological WHO subtypes, while distinct genetic alterations and molecular signatures directly influence their biological behavior and response to treatment. More recently, the incorporation of genomic analyses (i.e., DNA mutations, mRNA expression, and somatic copy number alterations) of TET patients’ data from The Cancer Genome Atlas (TCGA) identified four distinct molecular TET subtypes characterized by: (i) general transcription factor 2I (*GTF2I*) mutations, (ii) the overexpression of genes related to T cell signaling, (iii) chromosomal stability, and (iv) chromosomal instability. The cluster distinguished by the *GTF2I* mutations signature incorporates A and AB subtypes, while the T cell signaling cluster includes mainly B1 neoplasms and specifically correlates with T cell activation [2]. Further research integrating multi-omics data confirmed the enrichment of A and AB subtypes for *GTF2I* mutations, and additionally demonstrated a characteristic p53 loss-of-function and Myc overexpression for B and C TET subtypes [3]. The cornerstone of therapeutic protocols is surgical resection, with adjuvant/neoadjuvant treatment in more advanced stages. TETs exhibit a wide spectrum of clinical behaviors, with 30–40% of patients presenting co-existent autoimmune disorders, in particular thymoma-associated myasthenia gravis (TAMG) [4].

Marchevsky and Walts reported that programmed death-ligand 1 (PD-L1) is normally expressed in non-neoplastic thymi [5]. The programmed cell death protein 1 (PD-1)/PD-L1 interaction has been shown to regulate the positive selection of T cells in the thymus, thus shaping the final T cell repertoire [6,7]. So far, various studies have documented increased PD-L1 expression in TETs [5,8,9]. In the latest years, the introduction of immune checkpoint inhibitors (ICIs) has substantially prolonged patients’ overall survival (OS) in multiple solid and hematologic malignancies [10,11]. Their anti-tumor potential is based on tuning immunomodulatory activity. Extensive research from heterogeneous tumors has revealed several parameters that serve as determinants of patients’ response to immunotherapy with checkpoint blockades. The PD-L1 expression pattern on tumor cells and antigen-presenting cells (APCs) has been established as the most valuable predictor of patients’ response, which has also entered routine clinical practice [12]. Different PD-L1 expression levels, ranging from 1% to 50%, have been set as cut-offs for administering ICIs in different tumors [13]. For the case of TETs, phase I/II clinical trials assess the safety and efficacy of PD-1/PD-L1 inhibitors, namely, pembrolizumab, nivolumab, atezolizumab, and avelumab (reviewed in [14,15]), with pembrolizumab demonstrating encouraging results in the setting of relapsed and refractory cases [16,17]. The crucial role of PD-L1 expression profile as a distinctive factor between responders and non-responders to immunotherapy means that a better understanding of the molecular networks, especially the epigenetic programs governing PD-L1 transcription levels, could help identify novel candidate pathways of therapeutic intervention, which could be targeted in the setting of combination regimens to enhance the efficacy of immunotherapy.

Histone deacetylases (HDACs) are key epigenetic regulators of gene expression that remove the acetyl group from evolutionarily conserved lysine residues of histones, leading to chromatin condensation and subsequent transcriptional repression. Depending on their enzymatic activities and substrate specificities, they are classified into four classes, and their inhibition leads to a more open chromatin, allowing the access of transcription factors to regulatory sites, facilitating the activation of gene expression [18]. So far, studies have provided indirect evidence regarding the involvement of histone post-translational modifications, including deacetylation, in the pathogenesis of TETs (reviewed in [19]). Low *HDAC6* expression has been correlated with a dismal prognosis of patients with TETs based on the pan-cancer analysis of TCGA, genotype-tissue expression (GTEx), and the Cancer Cell Line Encyclopedia (CCLE) datasets [20]. Despite the fact that research has mainly focused on the effects of HDAC inhibitors on neoplastic cells, recent data support their potential to regulate the anti-tumor immune response via epigenetic modification, as well as their impact on the mechanisms mediating the effects of other anti-cancer therapies [21]. For the cases of melanoma and glioma, HDAC inhibitors have been shown to upregulate the expression of MHC-I and PD-1, and to affect PD-L1 expression via a STAT3-dependent mechanism, overall enhancing the immune response against the neoplastic cells and improving the effects of ICI therapy [22,23,24,25]. The efficacy of belinostat, an HDAC inhibitor, alone or combined with chemotherapy, has been evaluated in two phase II trials that included patients with advanced, recurrent, or refractory TETs [26,27]. Interestingly, belinostat showed an immunomodulatory activity leading a decrease in exhausted CD8+ T cells, a finding that correlated with its therapeutic efficacy [27]. Therefore, HDAC inhibitors, via their epigenetic reprogramming capacity, can alter the expression of immunomodulatory factors, including PD-L1, thus rendering their application a promising strategy to amplify the anti-tumor potential of PD-1/PD-L1 inhibitors, in the context of combination therapeutic regimens for TETs.

In this paper, we attempt to identify potential associations of PD-L1 expression patterns among different TETs histological types with the expression patterns of various isoforms of HDACs, investigating the possible roles of HDACs as predictors of immunotherapy response and as potential targets of combination schemes.

## 2. Materials and Methods

### 2.1. Study Population

This is a study of archival formalin-fixed paraffin-embedded (FFPE) tissue from 91 patients with TETs diagnosed in the period between 2009 and 2019, in the Department of Pathology in Evangelismos General Hospital Athens, Greece, for whom medical records were available. In total, 39 of the patients were men (43%) and 52 were women (57%), with a median age at diagnosis of 62 years (range 27–88 years). Gender was not correlated with WHO histological subtype or Masaoka–Koga stage. The frequencies of WHO TET subtypes were as follows: type A 13.2%, type AB 20.8%, type B1 15.4%, type B2 20.8%, type B3 15.4%, micronodular thymoma with lymphoid stroma (MNT) 2.2%, and type C 12.1%. Masaoka–Koga stage was I in 16.5%, IIa in 39.6%, IIb in 17.6%, III in 19.8%, IVa in 3.2%, and IVb in 3.2% of patients. Co-existing TAMG was diagnosed in 59.3% of patients, 2 of whom also suffered from pemphigus vulgaris and autoimmune thyroidopathy. Chemotherapy was given to 28% and radiotherapy to 50% of patients for whom respective information was available; 6 of these patients received both chemo- and radiotherapy. Follow-up information was available for 40 patients, with a follow-up duration ranging from 5 to 134 months (median: 32 months). There was not any significant association between gender and overall survival or time to relapse (*p* > 0.10). Patients and disease characteristics, as well as therapeutic modalities and outcomes, are summarized in Table 1.

### 2.2. Immunohistochemistry

Immunohistochemistry was carried out using standard procedures in the eight tissue microarrays (TMAs). Immunostainings for PD-L1 were performed on individual FFPE tissue sections, using anti-PD-L1 antibody SP142 C-terminal (Abcam, ab228462, Cambridge, MA, USA) and anti-PD-L1 SP263 (Dako, Agilent, Santa Clara, CA, USA).

In total, 87 of the samples were also stained for HDCA1, 70 for HDAC2, 79 for HDAC3, 86 for HDAC4, 82 for HDAC5 71 and for HDAC6, as described in our previous published study [28].

Antigen retrieval was performed at pH 6. The Envision (Dako, Agilent, Santa Clara, CA, USA) visualization system was used. DAB (3,3-diaminobenzidine) was used as a chromogen, and hematoxylin as a counterstain. Appropriate positive controls according to the manufacturer were used. The omitted primary antibody and substitution with an irrelevant antiserum were used in the negative control.

For the immunohistochemical evaluation, we calculated the H-score, which serves as a semiquantitative measure of the immunohistochemical proteins’ expression levels. To calculate the H-score, the semiquantitative staining intensity score (score 1 to 3) is multiplied by the percentage of positive cells. Therefore, H-score values range between 0 and 300 [29]. The epithelial and the lymphocytic components, as well as the nuclear and cytoplasmic positivity, were evaluated separately.

In the PDL-1 immunostained slides, two experienced pathologists evaluated the positive expression in the epithelial tumor cells as well as in the lymphocytes, and the tumor proportion score (TPS), combined proportion score (CPS) and immune cell score (IC-score) were calculated according to the published interpretation guidelines [30].

### 2.3. Statistical Analysis

Statistical analysis was performed by an MSc biostatistician (G.L.). The associations of the immunohistochemical expression of PDL-1 with both antibodies with clinicopathological characteristics as well as HDAC-1, -2, -3, -4, -5, and -6 expression were examined using non-parametric tests with correction for multiple comparisons, as appropriate. Survival analysis was performed using Kaplan–Meier survival curves, and the differences between the curves were compared with log-rank test. Numerical variables were categorized according to the median value. A *p*-value of <0.05 was considered statistically significant. The analysis was performed with the statistical package STATA 11.0/SE (College Station, TX, USA) for Windows.

## 3. Results

### 3.1. Expression of PD-L1 (SP263) in TETs

A positive expression of PD-L1 in the epithelial tumor cells was detected with SP263 in 49 of the examined cases (TPS ≥ 1%, 53.8%), showing complete or incomplete membranous staining. Representative images of the SP263 immunohistochemical staining in different TET subtypes are shown in Figure 1. Forty-nine cases also showed a positive stain in the lymphocytes. Seventeen cases without any staining in the epithelial tumor cells displayed a positive reaction in the tumor-infiltrating lymphocytes. CPS was positive in 78% of the cases (71/91). IC-score was calculated as 0 in 57% of the cases (52/91), 1 in 30.7% of the cases (28/91), 2 in 10.9% of the cases (10/91), and 3 in 1% of the cases (1/91).

B3 TETs and TCs showed a significantly higher PD-L1 (SP263) TPS expression compared to the rest of the TETs (Mann–Whitney U test, *p* = 0.006 for TPS, Figure 2). On the other hand, B3 TETs and TCs showed less frequently an IC-score > 0 (Fischer’s exact test, *p* = 0.033, 24% for B3 TETs and TCs versus 50% for the rest of the tumor types, Figure 3).

Moreover, advanced-stage tumors (III-IV) showed a higher PD-L1 (SP263) TPS expression compared to the low-stage ones (I-II) (Mann–Whitney U test, *p* = 0.037, Figure 2) and more frequently had a zero IC-score (Fischer’s exact test, *p* = 0.045, 72.7% for stage III/IV versus 45.8% for stage I/II, Figure 3).

There was no association between PD-L1 (SP263) expression and the presence of relapse or survival (*p* > 0.10).

### 3.2. Expression of PD-L1 (SP142) in TETs

A positive expression of PD-L1 in the epithelial tumor cells was detected with SP142 in 41 of the examined cases (TPS ≥ 1%, 45%), showing complete or incomplete membranous staining. Representative images of the SP142 immunohistochemical staining in different TET subtypes are shown in Figure 4. Seventy cases also showed a positive stain in the lymphocytes (76.9% of the cases). Thirty-eight cases without any staining in the epithelial tumor cells displayed a positive reaction in the tumor-infiltrating lymphocytes, and therefore CPS was positive in most of the cases (95.6%). The IC-score was calculated as 0 in 15.4% of the cases (14/91), 1 in 61.5% of the cases (56/91), 2 in 18.7% of the cases (17/91), and 3 in 4.4% of the cases (4/91).

PD-L1 (SP142) TPS expression was significantly higher in B3 TETs and TCs compared to the rest of the TET subtypes (Mann–Whitney U test, *p* = 0.0057 for TPS, Figure 5). Moreover, B3 TETs and TCs less frequently showed an IC-score > 0 compared to the rest of the TET subtypes; epithelial-rich TETs less frequently showed an IC-score > 0 (Fischer’s exact test, *p* = 0.018, 68% for B3 TETs and TCs versus 90.9% for the rest of the TET subtypes, Figure 6).

In addition, stage II-IV TETs showed a higher PD-L1 (SP142) TPS expression compared to the stage I TETs (Mann–Whitney U test, *p* = 0.001, Figure 5), while advanced-stage (III/IV) TETs were more frequently characterized by a zero IC-score compared to low-stage (I/II) TETs, although this correlation was of marginal significance (Fischer’s exact test, *p* = 0.083, 22.7% for stage III/IV versus 8.5% for stage I/II, Figure 6).

There was no statistically significant association between PD-L1 (SP263) expression and the presence of relapse or survival (*p* > 0.10).

### 3.3. Comparison of PD-L1 (SP263) and PD-L1 (SP142)

Six cases with a TPS < 1% assessed by PD-L1 (SP263) showed positive staining for PD-L1 in the epithelial tumor cells detected with SP142, ranging from 1 to 5%. On the other hand, fourteen cases with negative staining (TPS < 1%) for PD-L1 (SP142) displayed a positive reaction with PD-L1 (SP263), ranging from 1 to 60%. It should, however, be noticed that six of these cases showed a minimal positive expression with PD-L1 (SPS142), which was evaluated as less than 1% of the tumor cells.

Furthermore, 41 of the cases that did not have any positive staining in the lymphocytes (IC-score = 0) with PD-L1 (SP263) displayed a positive immunoreaction in these cells detected with the SP142 clone, mostly of score 1, with 1 case having a score of 3 and 7 cases having a score of 2. Three cases that did not have any positive staining in the lymphocytes (IC-score = 0) with PD-L1 (SP142), displayed a positive immunoreaction in these cells detected with the SP263 clone, of score 2 in two cases and of score 1 in one case. Table 2 summarizes the comparison of TPS positivity and IC-score values between PD-L1 (S263) and PD-L1 (SP142) staining.

### 3.4. Associations of PD-L1 Expression with HDAC-1, -2, -3, -4, -5 and -6

There was a positive correlation between PD-L1 (SP263) TPS and HDAC-5 cytoplasmic H-score, as well as HDAC-4 H-score (Spearman correlation coefficient rho = 0.31, *p* = 0.026 for HDAC-5 and rho = 0.36, *p* = 0.009 for HDAC-4).

Moreover, PD-L1 (SP142) TPS was positively correlated with HDAC-3 cytoplasmic H-score, HDAC-5 cytoplasmic H-score as well as HDAC-4 H-score (Spearman correlation coefficient rho = 0.30, *p* = 0.034 for HDAC-3, rho = 0.47, *p* < 0.001 for HDAC-5 and rho = 0.33, *p* = 0.016 for HDAC-4). Table 3 summarizes the associations of PD-L1 (SP263) and PD-L1 (SP142) TPS with the expressions of the different HDACs studied.

## 4. Discussion

PD-L1 is expressed in both antigen-presenting cells (APCs) and neoplastic cells, and acts as a brake, interfering with the effective anti-tumor immune response at two levels. First, it negatively regulates the interplay between T cells and APCs, dampening the effective priming of the former, thus hindering a potent tumoricidal adaptive immune response. Second, PD-L1 expression on various tumor cells induces inhibitory signals after ligation to PD-1 expressed on cytotoxic T-cells, downregulating antitumor immunity and allowing neoplastic cells to evade immunosurveillance [31]. Consequently, the effectiveness of the PD-L1 blockade in cancer immunotherapy depends not only on the level of PD-L1 expression by tumor cells, but also on its expression by host immune cells [32]. Therefore, in routine clinical practice, the evaluation of PD-L1 expression is determined in both tumor and immune cells, with various cut-off limits, ranging from ≥1% to ≥50% [33]. In the current study, we evaluated PD-L1 expression in a large cohort of TETs with two different clones (SP142, SP263), both in tumor and immune cells, and investigated its correlation with tumor subtypes, disease stage, relapse, and survival. We also explored the possible correlation of PD-L1 expression with various HDAC isoforms’ expression.

Based on our results, B3 TETs and TCs demonstrate statistically significantly higher PD-L1 TPS compared to other TET subtypes assessed with either SP263 or SP142. For SP263, tumors of an advanced stage (III–IV) showed a statistically significantly higher PD-L1 TPS compared to those of a low stage (I–II), while for SP142, a statistically significantly higher expression was observed for stages II–IV compared to stage I. Our findings are in accordance with previously published retrospective studies analyzing PD-L1 via immunohistochemical assays, which have demonstrated specific patterns regarding its expression profile among distinct WHO histological subtypes and Masaoka–Koga pathological stages. In most of these studies, higher PD-L1 immunohistochemical levels were detected in more aggressive WHO subtypes (i.e., B2, B3, and C) and were associated with more advanced Masaoka–Koga stages [8,34,35,36,37,38]. In one of these studies, PD-L1 mRNA levels were also assessed by qPCR, presenting an increase relevant to that of the protein levels, with an occasional association with PD-L1 gene amplification [37].

Another interesting finding of our study is that B3 TETs and TCs were less frequently characterized by PD-L1 IC-score > 0, namely, 24% versus 50% for the rest of the TET subtypes with SP263 staining, and 68% versus 90.9% for the rest of the TET subtypes with SP142 staining. In several other types of cancer PD-L1, expression on tumor-infiltrating lymphocytes has been associated with favorable prognosis [39,40]. In this context, it has been shown that PD-L1 upregulation is essential for effector T cells to survive during the contraction phase of the immune response and to elicit protective immunity [41].

In the present study, no statistically significant correlations were documented between the expression of PD-L1 and the relapse rates or OS regardless of the PD-L1 clone used for immunohistochemical detection. Based on the published literature, contradictory results have been derived regarding the prognostic value of PD-L1, as it has been correlated with both better and worse OS and disease-free survival (DFS). Such differences could be attributed mainly to two factors: the different cut-off limits used among studies, and the variations detected among distinct clones of anti-PD-L1 antibodies [42]. As for the latter, we should note that differences in the expression levels of PD-L1 both on epithelial tumor cells and immune cells are also seen in our study depending on the clone of anti-PD-L1 used. Namely, 14 out of the 91 cases with negative PD-L1 expression (<1%) on epithelial tumor cells with the SP142 clone were characterized by positive expression when examined with the SP246 clone. We should highlight that this positivity was not marginal, given that its levels ranged between 1% and 60%. Interestingly, when it comes to the PD-L1 IC-score, it seems that staining with the SP142 clone yielded positivity for as many as 41 cases that were characterized by a zero IC-score with the SP263 clone. These findings might suggest a better sensitivity of the SP246 clone for PD-L1 detection on tumor epithelial cells and of the SP142 clone for PD-L1 detection on immune cells of TETs. Undoubtedly, further investigations are required to validate this observation.

Given that most of the relative research, including ours, reports the levels of PD-L1 expression at TET diagnosis, we should highlight the dynamic and multi-factorial nature of PD-L1 regulation in the course of the disease. Intriguing findings have been extracted from two studies that retrospectively analyzed specimens of primary and metastatic/recurrent lesions, as well as of pre- and post-chemotherapy specimens. In the first one, Terra et al., using two cut-offs for PD-L1 positivity (i.e., 1% and 50%), highlighted a deviation of PD-L1 expression in up to 20% of cases between primary tumors and recurrences or metastases, with no administration of systemic treatment (radiotherapy or chemotherapy) between the two timepoints of specimen collection [43]. The second study, attempting a comparison of PD-L1 and PD-1 expression profiles pre- and post-chemotherapy, showcased the statistically significant upregulation of PD-L1 after neoadjuvant chemotherapy and radiation therapy. On the other hand, while an increase was also encountered in the expression of PD-1 on immune cells, no statistically significant correlation was established [44].

The above findings, which demonstrate substantial heterogeneity in the expression patterns of PD-L1, depending on the tumor phenotype and pathological stage, as well as on previous treatment administration, indicate the complex nature of the regulatory networks governing PD-L1 expression. PD-L1 expression by tumor cells is controlled by both intrinsic and microenvironmental factors. The activation of oncogenic signaling cascades, gene amplification, and disruption of the gene regulation networks are implicated in PD-L1 upregulation in tumor cells [45]. Moreover, the continuous crosstalk of neoplastic cells with TME, mediated via a plethora of inflammatory cytokines such as interferon-γ (IFNγ) and tumor necrosis factor α (TNF-α), triggers various intracellular pathways that lead to the transcriptional activation of PD-L1 [46,47]. The heterogeneous signaling pathways controlling PD-L1 expression converge to several transcription factors and epigenetic modifiers [48]. The latter coordinate the reorganizing of chromatin structure into a more permissive state and enable the former to find access to regulatory regions and prompt gene expression. The specificity of factors involved in PD-L1 regulation varies among various types of cells in a context-dependent manner. In the case of thymic neoplasms, higher PD-L1 expression is detected in tumors bearing *TP53* and *PTEN* alterations [49]. A study aiming to provide insight into the molecular pathways regulating PD-L1 expression was conducted on four TET cell lines, and identified the cylindromatosis (*CYLD*) gene as a crucial regulator [50]. A significant association was observed between low *CYLD* expression and ≥50% PD-L1 expression, while *CYLD* knockdown led to the upregulation of IFN-γ-mediated activation of the signal transducer and activator of transcription 1 (STAT1)/IFN regulatory factor 1 (IRF1) axis, which in turn upregulated PD-L1 expression [50].

Given the lack of data concerning the effects of epigenetic modifiers on the control of PD-L1 in the setting of TETs, in the current study, we tried to clarify the potential association between HDACs and PD-L1 expression, as similar correlations have been encountered in a broad spectrum of hematologic and solid neoplasms. Cumulative data suggest that PD-L1 regulation is exerted mainly by class I and class II HDACs. In a cohort of human non-small cell lung cancer (NSCLC) specimens, HDAC10 (class II) expression positively correlated with PD-L1, with high-level expression of HDAC10 indicating dismal patient prognosis [51], while in hepatocellular carcinoma samples, increased HDAC9 (class II) and HDAC2 (class I) expression showed a positive correlation with PD-L1 levels [52]. In melanoma, PD-L1 seems to be directly controlled primarily by class I HDACs (HDAC1, HDAC2, HDAC3, HDAC8) [23]. In the present study, the evaluation of PD-L1 expression in TETs with the SP263 clone highlighted a positive correlation between PD-L1 (SP263) TPS and the cytoplasmic H-scores of HDAC-5 and HDAC-4. Interestingly, the evaluation of PD-L1 expression with the SP263 clone not only confirmed the positive correlation between PD-L1 expression and the cytoplasmic H-scores of HDAC-5 and HDAC-4, but also revealed a positive correlation, though less significant, with HDAC-3 cytoplasmic H-score.

Strikingly, our results point to correlations of certain HDACs cytoplasmic H-scores and PD-L1 expression, while no significant correlations could be deduced for any of the HDACs’ nuclear H-scores. Despite the well-established role of HDACs in epigenetic regulation via histone deacetylation in the nucleus, the HDAC-mediated deacetylation of lysine residues on transcription factors and key proteins of signaling pathways in the cytoplasm can indirectly control the transcriptional regulation of certain genes, including PD-L1. The correlation of PD-L1 expression and cytoplasmic detection of HDAC4 and HDAC5 reported in this study complies with the function of class II HDACs, which mainly comprises the regulation of non-histone proteins. Though the exact pathways leading to PD-L1 regulation by class II HDACs have not been fully elucidated, data derived from studies in pancreatic cancer demonstrate that HDAC5 negatively regulates PD-L1 expression via interaction with NF-κΒ p65. Zhou et al. reported that HDAC5 diminished p65 acetylation at lysine-310, which is essential for the transcriptional activity of p65. HDAC5 inhibition led to increased PD-L1 expression, and thus sensitized tumor cells to immune checkpoint blockade [53].

For the case of the class I HDAC3, for which we documented a correlation of its cytoplasmic localization and PD-L1, despite its main function at the level of histone deacetylation in the nucleus, research supports that it can also act in the cytoplasm. Though, to our knowledge, there are no studies directly correlating HDAC3 cytoplasmic function and PD-L1, it seems that HDAC3 regulates the regulators of PD-L1 expression. It has been widely established that the upregulation of PD-L1 in cancer cells is controlled via NF-κB [54]. HDAC3-induced deacetylation of the RelA subunit of NF-κB mediates the nuclear export of NF-κΒ, thus controlling the duration of NF-κB transcriptional response [55]. Moreover, HDAC3 potently deacetylates the p65 subunit of NF-κΒ, downregulating its transcriptional activity [56]. A phosphor-acetyl switch has been shown to regulate STAT1 signaling, since STAT1 acetylation leads to its decreased phosphorylation, therefore resulting in its reduced nuclear translocation. HDAC3 is responsible for STAT1 deacetylation and subsequently increased nuclear translocation [57]. Moreover, the acetylation of STAT family proteins on Lys685 residue is critical for the formation of stable STAT dimers, but can be reversed by type I HDACs [58]. PD-L1 expression can be increased by IFN-γ via JAK2/STAT1 signaling [59]. We can therefore presume that the HDAC3-mediated deacetylation of STAT1 can indirectly contribute to increased PD-L1 expression via the enhanced dimerization and nuclear translocation of STAT1.

HDAC inhibition has been shown to upregulate PD-L1 expression in several tumor types. More specifically, the targeted pharmacological inhibition of class I HDAC isoforms, especially of HDAC1, HDAC2, HDAC3, and HDAC8 on melanoma cell lines, prompted an upregulation of PD-L1 and PD-L2 on tumor cells by increasing the acetylation levels of their regulatory regions, leading to chromatin relaxation, and the enhancement of gene expression [23]. In multiple myeloma cell lines, the pan-HDACi panobinostat has demonstrated the capacity to markedly upregulate PD-L1 by enhancing STAT1 expression in the presence of IFN-γ [60], while the HDAC6-selective inhibitor A452 was shown to induce PD-L1 upregulation in multiple myeloma cells in vitro [61]. The administration of broad-specificity HDACi, targeting class I and II isoforms, has been shown to induce an increase in PD-L1 expression in anaplastic thyroid carcinoma cells [62], while class I-targeting agents have displayed significant potential to upregulate PD-L1 in triple-negative breast carcinoma, head and neck squamous cell carcinoma, NSCLC, and Hodgkin lymphoma [63,64,65,66,67]. In bladder urothelial carcinoma and chronic lymphocytic leukemia (CLL) cell lines, selective HDAC6 blockade prompted PD-L1 upregulation [68,69]. These observations have led to the use of a combination of HDACi and PD-L1 inhibitors for the treatment of certain neoplasms, given that the increase in PD-L1 expression renders anti-PD-L1 treatment more effective [61,70,71]. The correlations between the expression of PD-L1 and different HDACs in TETs, which are described for the first time in our study, could be exploited for a more effective therapeutic targeting of these tumors.

The therapeutic potential of anti-PD-1/PD-L1 agents lies in their ability to reprogram the host immune system to propagate a robust effector adaptive immune response against neoplastic cells. Besides the expression of immune checkpoint molecules on tumor cells, the presence of an immunologically “hot” TME, infiltrated by cytotoxic T cells, and the tumor mutational burden (TMB) that provides adaptive immunity with neo-antigen targets against which a robust cell-mediated response can arise, are also pre-requisites for the effectiveness of anti-PD-1/PD-L1 therapeutic targeting. Regarding TMB in TETs, it is, as expected, directly related to the tumor phenotype, as TCs are characterized by a significantly larger load of mutant epitopes (*p* < 0.001), while advanced Masaoka–Koga stages (III-IV) are also defined by higher mutational burden (*p* < 0.001) [72]. TMB is also a determining factor affecting the composition of immune infiltrates, such as activated NK cells, macrophages, resting mast cells, activated mast cells, neutrophils, T cells CD4 naive, regulatory T cells, naive B cells, and B cell memory between the low- and high-TMB groups.

To conclude, regarding PD-L1 expression in TETs studied by immunohistochemistry, we should note that, to our knowledge, our research includes one of the largest published cohorts (91 patients), assessing PD-L1 expression in tumor cells and infiltrating immune cells with two different PD-L1 antibody clones. As for the two other published cohorts including a slightly higher number of patients, Weissferdt et al. presented results of 98 TET samples stained with one PD-L1 antibody clone, reporting PD-L1 expression only on tumor cells [9], while Rouquette et al. assessed PD-L1 expression both on tumor and immune cells in 103 TET samples, testing four PD-L1 antibody clones [8]. The novelty of the present study lies in the combined evaluation of PD-L1 and different HDAC subtypes in TETs not previously described in the literature, which can lead to further investigations of the pathways implicated in PD-L1 regulation by HDACs in TETs, and most importantly, offer insights into new therapeutic targeting approaches.

## 5. Conclusions

Our study highlights that the more aggressive TET subtypes, as well as advanced-stage TETs, are characterized by increased PD-L1 expression in neoplastic cells and decreased PD-L1 expression in the immune cells of the TME. Overall, these findings mirror the dampening of anti-tumor immunity induced by PD-L1 expression on tumor cells, and the enhancement of anti-tumor properties when PD-L1 is expressed on immune cells. The novelty of this study lies in the observed positive correlation between PD-L1 and certain HDAC expression. The epigenetic modifications induced by HDACs are known to mediate the regulation of PD-L1 expression. In the setting of TETs, this is the first relevant report that not only contributes to deciphering the pathways regulating PD-L1 expression, but also unveils perspectives for combinational therapeutic targeting.

## Figures and Tables

**Figure 1 biomedicines-12-00772-f001:**
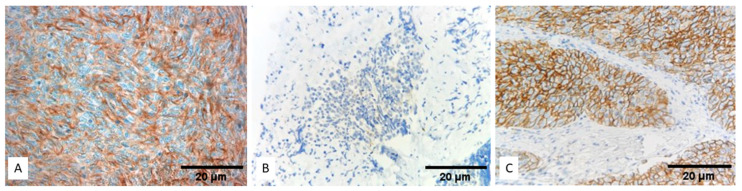
Immunohistochemical expression of PD-L1 with SP263 in different TET subtypes. (**A**) TET subtype A, (**B**) TET subtype B2, (**C**) TET subtype B3 (Magnification ×40).

**Figure 2 biomedicines-12-00772-f002:**
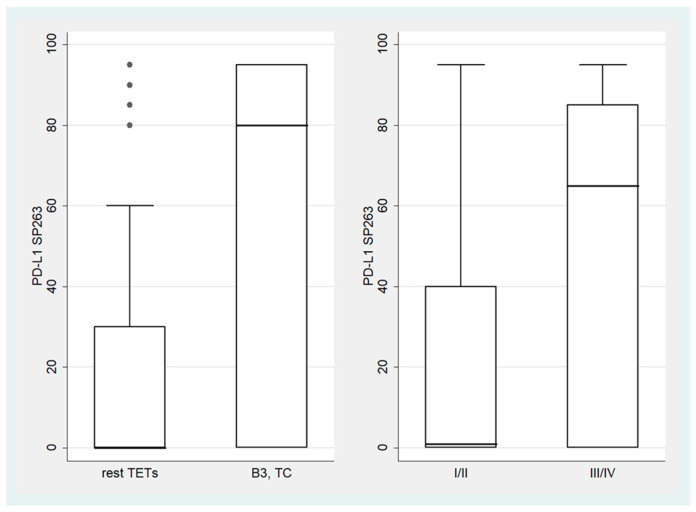
Schematic representation of the association between PD-L1 (SP263) TPS and WHO TET subtypes (**left panel**), as well as Masaoka–Koga stage (**right panel**). Dots represent outliers. TC, thymic carcinoma; TETs, thymic epithelial tumors.

**Figure 3 biomedicines-12-00772-f003:**
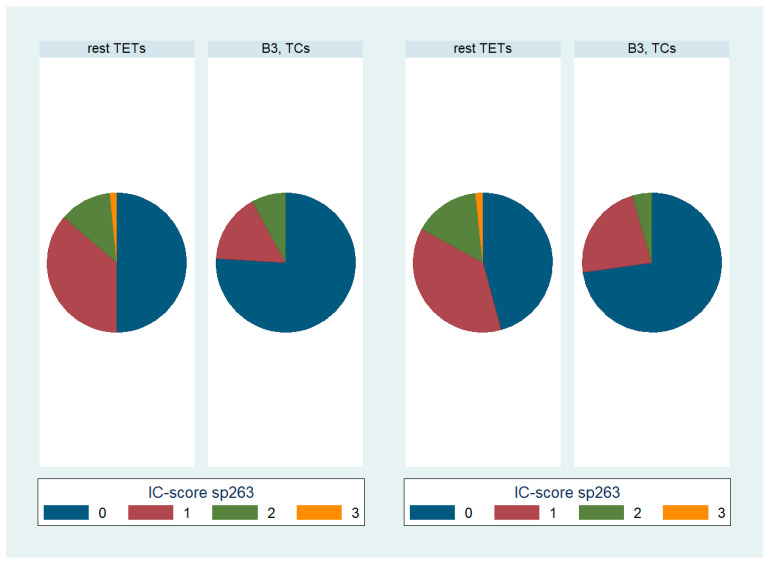
Schematic representation of the distributions of various PD-L1 (SP263) IC-score categories among WHO TET subtypes (**left panel**), as well as Masaoka–Koga stage (**right panel**). IC-score, immune cell score; TC, thymic carcinoma; TETs, thymic epithelial tumors.

**Figure 4 biomedicines-12-00772-f004:**
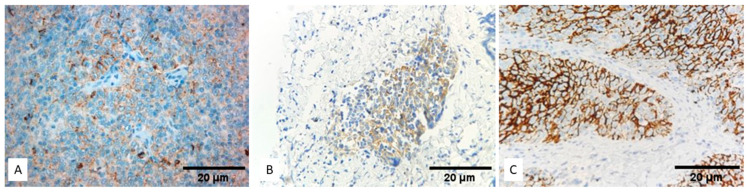
Immunohistochemical expression of PD-L1 with SP142 in different TET subtypes. (**A**) TET subtype A, (**B**) TET subtype B2, (**C**) TET subtype B3 (Magnification ×40).

**Figure 5 biomedicines-12-00772-f005:**
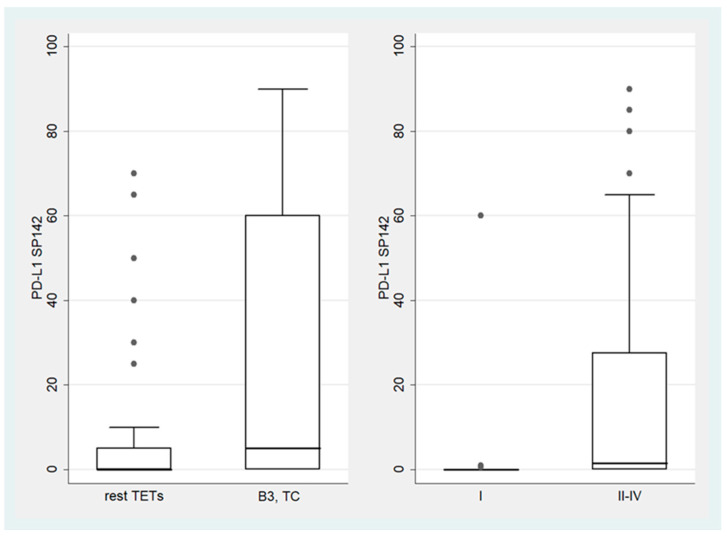
Schematic representation of the association between PD-L1 (SP142) TPS and WHO TET subtypes (**left panel**), as well as Masaoka–Koga stage (**right panel**). Dots represent outliers. TC, thymic carcinoma; TETs, thymic epithelial tumors.

**Figure 6 biomedicines-12-00772-f006:**
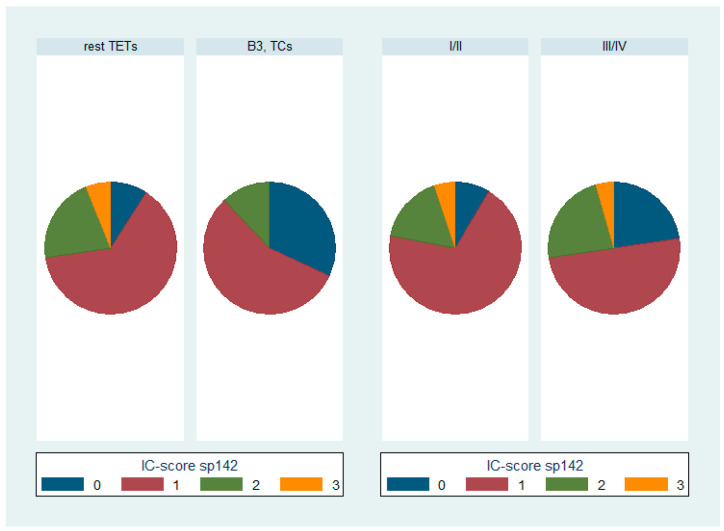
Schematic representation of the distributions of various PD-L1 (SP142) IC-score categories among WHO TET subtypes (**left panel**), as well as Masaoka–Koga stage (**right panel**). IC-score, immune cell score; TC, thymic carcinoma; TETs, thymic epithelial tumors.

**Table 1 biomedicines-12-00772-t001:** Clinicopathological characteristics of 91 patients with TETs.

Parameter	Median	Range
Age (years)	62	27–88
	**Number**	**%**
**Gender**		
Male	39/91	43%
Female	52/91	57%
**WHO subtypes**		
Type A	12/91	13.6%
Type AB	19/91	20.8%
Type B1	14/91	15.4%
Type B2	19/91	20.8%
Type B3	14/91	15.4%
Micronodular with lymphoid stroma	2/91	2.2%
Type C	11/91	12.1%
**Masaoka–Koga stage**		
I	15/91	16.5%
IIa	36/91	39.6%
IIb	16/91	17.6%
III	18/91	19.8%
IVa	3/91	3.2%
IVb	3/91	3.2%
**TAMG**	35/59	59.3%
**Adjuvant/neoadjuvant treatment**		
Chemotherapy	11/39	28%
Radiotherapy	19/38	50%
**Survival outcomes**		
Alive	29/40, follow-up 5–134 months	72.5%
Dead of disease	11/40, within 7–65 months	27.5%
**Relapse**	4/35, within 58–65 months	11%

TAMG, thymoma-associated myasthenia gravis.

**Table 2 biomedicines-12-00772-t002:** Comparison of TPS positivity and IC-score values between PD-L1 (SP263) and PD-L1 (SP142) staining.

	PD-L1 (SP263)	PD-L1 (SP142)
TPS < 1%	49/91 (53.8%)	41/91 (45%)
TPS ≥ 1%	42/91 (46.2%)	50/91 (55%)
IC-score = 0	52/91 (57.0%)	14/91 (15.4%)
IC-score ≥ 1	39/91 (43.0%)	77/91 (84.6%)

**Table 3 biomedicines-12-00772-t003:** Associations between HDAC-1, -2, -3, -4, -5 and -6 H-score with PDL-1 (SP142) and PDL-1 (SP263) according to Spearman’s correlation coefficient. Statistically significant results are marked with a star.

	PD-L1 (SP142) TPS	PD-L1 (SP263) TPS
Nuclear HDAC-1 H-score	rho = 0.01	rho = −0.08
	*p* = 0.954	*p* = 0.597
Cytoplasmic HDAC-1 H-score	rho = 0.06	rho = −0.05
	*p* = 0.684	*p* = 0.735
HDAC-2 H-score	rho = 0.24	rho = 0.18
	*p* = 0.093	*p* = 0.408
Nuclear HDAC-3 H-score	rho = −0.04	rho = −0.07
	*p* = 0.768	*p* = 0.627
Cytoplasmic HDAC-3 H-score	rho = 0.32	rho = 0.30
	*p* = 0.091	***p* = 0.034 ***
HDAC-4 H-score	rho = 0.36	rho = 0.33
	***p* = 0.009 ***	***p* = 0.016 ***
Nuclear HDAC-5 H-score	rho = 0.19	rho = 0.27
	*p* = 0.170	*p* = 0.0508
Cytoplasmic HDAC-5 H-score	rho = 0.31	rho = 0.47
	*p* = 0.026 *	***p* < 0.001 ***
HDAC-6 H-score	rho = 0.02	rho = 0.16
	*p* = 0.907	*p* = 0.253

## Data Availability

The data presented in this study are available on request from the corresponding author.

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
