# Peer review of "PD-L1 Expression in Neoplastic and Immune Cells of Thymic Epithelial Tumors: Correlations with Disease Characteristics and HDAC Expression"

_biomedicines, 2024, doi:10.3390/biomedicines12040772_

Round 1

Reviewer 1 Report

Comments and Suggestions for Authors

General comments:

The authors found that Higher PD-L1 expression in neoplastic cells and lower PD-L1 expression in immune cells of TETs characterizes more aggressive and advanced neoplasms. The data were based on the archival formalin-fixed paraffin-embedded (FFPE) tissue from 95 TETs diagnosed patients. This manuscript Our study highlights that the more aggressive TET subtypes as well as advanced-stage TETs are characterized by increased PD-L1 expression in neoplastic cells and decreased PD-L1 expression in the immune cells of the TME.

Minor comments:

1. p value: p should be typed in italic font.

2. Data sources for 3.3. Comparison of PD-L1 (SP263) and PD-L1 (SP142) and 3.4. Associations of PD-L1 expression with HDAC-1, -2, -3, -4, -5 and -6 need to mention. If this is a new data, please consider to add the Table in addition to result description.

Author Response

We sincerely appreciate your valuable feedback and thank you for your comments.

Below, we address each of your comments in a point-by-point manner.

Comment 1: “p value: p should be typed in italic font”

Response to Comment 1: Relevant changes have been made.

Comment 2: “Data sources for 3.3. Comparison of PD-L1 (SP263) and PD-L1 (SP142) and 3.4. Associations of PD-L1 expression with HDAC-1, -2, -3, -4, -5 and -6 need to mention. If this is a new data, please consider to add the Table in addition to result description.”

Response to Comment 2:  We would like to thank you for the insightful recommendation. We have added two tables, “Table 2” for the results described in section 3.3. and “Table 3” for the results described in section 3.4.

Reviewer 2 Report

Comments and Suggestions for Authors

In this study, the authors attempted to identify potential associations of PD-L1 expression patterns among different thymic epithelial tumors (TETs) histological types with the expression patterns of various isoforms of HDACs, investigating the possible role of HDACs as predictors of immunotherapy response and as potential targets of combination schemes. The authors concluded that higher PD-L1 expression in neoplastic cells and lower PD-L1 expression in immune cells of TETs characterizes more aggressive and advanced neoplasms. Correlations between PD-L1 and HDAC expression unravel the impact of epigenetic regulation on the expression of immune checkpoint molecules in TETs, with possible future applications in combined therapeutic targeting.

Comments:

The reviewer has some concerns as follows:

1.     The roles of PD-L1 and HDACs in the TETs or other cancers has been shown or reviewed, such as Dapergola et al. Emerging therapies in thymic epithelial tumors (Review). Oncol Lett. 2023 Jan 16;25(2):84.; Agrafiotis et al. Immunotherapy and Targeted Therapies Efficacy in Thymic Epithelial Tumors: A Systematic Review. Biomedicines. 2023 Oct 8;11(10):2722.; Nicolì V, Coppedè F. Epigenetics of Thymic Epithelial Tumors. Cancers (Basel). 2023 Jan 5;15(2):360.; Moran et al. The impact of histone deacetylase inhibitors on immune cells and implications for cancer therapy. Cancer Lett. 2023 Apr 10;559:216121. These information need to be added in the introduction of this manuscript.

2.     Is there gender difference for survival outcomes shown in Table 1?

3.     In Figures 1 and 4, the scale bars need to be added.

4.     In Figures 2 and 5, what do the black dots on the figure mean?

5.     Please provide the figures for associations of PD-L1 expression with HDACs.

Author Response

We would like to express our sincere gratitude for your diligent review of our manuscript. Your insightful comments and suggestions have been invaluable in refining the quality of our work. Below, we address each of your comments in a point-by-point manner.

Comment 1: “The roles of PD-L1 and HDACs in the TETs or other cancers has been shown or reviewed, such as Dapergola et al. Emerging therapies in thymic epithelial tumors (Review). Oncol Lett. 2023 Jan 16;25(2):84.; Agrafiotis et al. Immunotherapy and Targeted Therapies Efficacy in Thymic Epithelial Tumors: A Systematic Review. Biomedicines. 2023 Oct 8;11(10):2722.; Nicolì V, Coppedè F. Epigenetics of Thymic Epithelial Tumors. Cancers (Basel). 2023 Jan 5;15(2):360.; Moran et al. The impact of histone deacetylase inhibitors on immune cells and implications for cancer therapy. Cancer Lett. 2023 Apr 10;559:216121. These information need to be added in the introduction of this manuscript.”

Response to Comment 1: We would like to thank you for this valuable suggestion. The inclusion of the information presented in the literature you have suggested contributes to a better understanding of the role of PD-L1 and HDACs in TETs’ pathogenesis and therapeutics.  This information has been added in the introduction, along with further relevant literature references, in lines 64-68, lines 77-81, and lines 92-108.

Comment 2: “Is there gender difference for survival outcomes shown in Table 1?”

Response to Comment 2: There was not any significant association between gender and overall survival or time to relapse (p>0,10). Moreover, gender was not correlated with WHO histological subtype or Masaoka-Koga stage. This information has been included in the manuscript lines 123-124 and lines 133-134.

Comment 3: “ In Figures 1 and 4, the scale bars need to be added.”

Response to Comment 3: Scales have been added to the figures.

Comment 4: “In Figures 2 and 5, what do the black dots on the figure mean?”

Response to Comment 4: Dots represent outliers (values that fall above or below the end of the whiskers and are plotted as dots). We have included this clarification in the figure legends.

Comment 5: “ Please provide the figures for associations of PD-L1 expression with HDACs.”

Response to Comment 5: We would like to thank you for your attention to detail and your valuable suggestion. We found it more appropriate to include a Table (Table 3 in the revised manuscript) describing these correlations, according to the suggestions of Reviewer 1.

Round 2

Reviewer 2 Report

Comments and Suggestions for Authors

This revised manuscript has a great improvement and can be accepted.

Author Response

We would like to thank the reviewer for the contribution to refine the quality of our manuscript through the insightful comments made in the first review round.

We are glad we have adequately answered your comments.